# Virus-Associated Nephropathies: A Narrative Review

**DOI:** 10.3390/ijms231912014

**Published:** 2022-10-10

**Authors:** Christophe Masset, Paul Le Turnier, Céline Bressollette-Bodin, Karine Renaudin, François Raffi, Jacques Dantal

**Affiliations:** 1Institut de Transplantation Urologie Néphrologie (ITUN), Service de Néphrologie et Immunologie Clinique, CHU Nantes, 44000 Nantes, France; 2Center for Research in Transplantation and Translational Immunology, Nantes Université, INSERM, UMR 1064, 44000 Nantes, France; 3Infectious Diseases Department, Cayenne Hospital, Cayenne 97300, French Guiana; 4Laboratoire de Virologie, CHU de Nantes, 44000 Nantes, France; 5Service d’Anatomie et de Cytologie Pathologique, CHU de Nantes, 44000 Nantes, France; 6Department of Infectious Diseases, University Hospital of Nantes, and CIC 1413 INSERM, 44000 Nantes, France

**Keywords:** viral infection, nephropathy, acute kidney injury, glomerulopathy

## Abstract

While most viral infections cause mild symptoms and a spontaneous favorable resolution, some can lead to severe or protracted manifestations, specifically in immunocompromised hosts. Kidney injuries related to viral infections may have multiple causes related to the infection severity, drug toxicity or direct or indirect viral-associated nephropathy. We review here the described virus-associated nephropathies in order to guide diagnosis strategies and treatments in cases of acute kidney injury (AKI) occurring concomitantly with a viral infection. The occurrence of virus-associated nephropathy depends on multiple factors: the local epidemiology of the virus, its ability to infect renal cells and the patient’s underlying immune response, which varies with the state of immunosuppression. Clear comprehension of pathophysiological mechanisms associated with a summary of described direct and indirect injuries should help physicians to diagnose and treat viral associated nephropathies.

## 1. Introduction

Viruses are among the most common pathogens involved in infectious diseases and have a global distribution [1]. While most viral infections cause mild symptoms and spontaneously resolve, some can lead to severe or protracted manifestations, specifically in immunocompromised hosts, underlying the usefulness of antiviral therapies and the need for specific vaccines [2,3].

Viral infections represent an important consideration for healthcare workers in charge of renal diseases. First, patients with kidney diseases are often immunocompromised (treatments of autoimmune diseases, patients with end-stage renal disease requiring dialysis, kidney transplant recipients—KTRs) and may be confronted with opportunistic infections, including those caused by viruses [4,5]. Second, viral infections may be difficult to manage in these patients due to poorer responses to vaccination and possible nephrotoxicity induced by antiviral drugs [6]. Finally, viruses can directly or indirectly affect the kidneys, resulting in a wide range of kidney diseases mediated by various mechanisms, the most common being either a direct tropism of the virus for the kidney tissue or the development of immune complexes triggered by the pathogen [7,8,9].

The literature on kidney injuries related to viral infections is broad due to the multiplicity of potentially involved viruses and the numerous causes: hypovolemia secondary to infection severity, direct or indirect viral-associated nephropathy and drug toxicity being the major ones. The aim of this review is to provide physicians with broad but clear information about virus-associated nephropathies in order to guide diagnosis strategies and treatments in cases of acute kidney injury (AKI) occurring concomitantly with a viral infection.

## 2. Human Immunodeficiency Virus (HIV)

The HIV/AIDS pandemic remains a worldwide health issue. There have been about 33 million deaths in the past 40 years, and there are 38 million people living with HIV [10]. Numerous HIV-related renal diseases have been described, and these are linked either to the direct action of the virus or to an inappropriate immune response. It should be noted that impaired renal function in these patients may be related to opportunistic infections or antiretroviral therapy side-effects, which have been reviewed extensively elsewhere [11]. Glomerular disease is the predominant representation of HIV-related renal injury, particularly through HIV-associated nephropathy (HIVAN), which occurs at high viral loads and low CD4^+^ T-cell counts [12]. Its incidence has largely decreased with the efficacy and standardization of combination antiretroviral therapy (cART) [13]. Glomerular involvement classically associates a collapsed focal segmental glomerulosclerosis (FSGS) pattern with microcystic dilatation of the renal tubules and interstitial inflammation [14]. The pathogenicity of HIVAN has been extensively studied and includes both direct and indirect pathways [15]. In vitro models established that infection mechanisms of renal cells by HIV differ from those usually described in CD4^+^ T cells. Indeed, DC-SIGN (CD 209) was a preferential receptor for podocyte infection by HIV, and DEC-205 was the main one for tubular epithelial cells [16]. Additionally, entry of HIV proteins (notably Tat and Vpr) in podocytes can occur through heparan sulfate and lipids rafts, which seems to induce further evolution toward podocytes’ apoptosis [17]. Thus, HIV can directly impact podocytes (and also tubular cells), resulting in dedifferentiation, cell proliferation and finally apoptosis mediated by expression of HIV transgenes [18]; see Figure 1a. HIV-induced interferon expression may be responsible for additional podocyte toxicity, especially in patients carrying *APOL-1* at-risk alleles [19]. For this reason, African American and sub-Saharan populations are at particular risk for HIVAN because they harbor specific *APOL-1* genotype variations [20].

HIV immune-complex renal disease (HIVICD) is nowadays the leading glomerulopathy in the era of cART [13]. Indeed, HIVICD is an umbrella term that encompasses a large complex of renal injuries due to the anti-HIV immune response, promoted by the immune reconstitution induced by cART [21]. Once again, glomerular damage is the most frequently observed pattern. HIV-specific immune complexes are deposited in the glomerular tuft, resulting in endothelial damage, complement activation and immune cell infiltration and proliferation [21]. Nevertheless, whilst HIV antigens (p24 and gp120) within immune complexes isolated from kidney biopsies of patients suffering from IgA nephropathy have been observed, the direct relation between HIV infection and glomerulonephritis remains debated. In addition, membranous nephropathy and membranoproliferative glomerulonephritis often occur in the setting in of coinfection with hepatitis B and/or C, or even following a bacterial infection [22]. Finally, the reversibility of kidney impairment with HIV treatment in these nephropathies is inconsistent, and the place for anti-inflammatory agents remains debated [23].

Diffuse infiltrative lymphocytosis syndrome (DILS) is another presentation of HIV-related nephropathies, represented by a tubulointerstitial infiltrate of polyclonal reactive cells [24]. Finally, although rare, local HIV infection of endothelial cells can result in damage that increases the risk of thrombotic microangiopathy (TMA), especially if patients carry other risk factors [25,26].

## 3. Hepatitis Viruses 

### 3.1. Hepatitis A Virus (HAV)

HAV is a nonenveloped single-stranded RNA virus belonging to the *Picornaviridae* family that approximately infects 1.5 million people each year [27]. HAV-induced hepatocellular apoptosis and inflammation have been associated with the innate immune response [28]. Acute HAV infection usually presents as a self-limited illness, indistinguishable from other types of acute viral hepatitis. Extra-hepatic manifestations rarely occur, but AKI may be present in 5% of patients [29,30]. AKI may also be present in patients with non-fulminant hepatitis A [29,31]. Reported renal injuries are mostly acute tubular necrosis (ATN) [29], but cases of acute tubulo-interstitial nephritis (AITN) [31,32] and glomerular nephritis [33,34] have been described. To penetrate host cells, HAV mainly binds to TIM1 [35,36] (T cell immunoglobulin mucin domain 1), also known as KIM1 (kidney injury molecule 1), which can be highly expressed by kidney cells. However, HAV has not been detected in renal tissue to directly cause renal damages. The outcomes are mostly good, even for severe AKI [31]. A higher risk of severe disease in renal transplant recipients has been suggested by some authors, but the data remain too scarce to be conclusive [37]. No anti-HAV specific therapy exists. Hepatitis A vaccination is highly effective [38], and the usual two-doses scheme is effective in kidney transplant recipients [39].

### 3.2. Hepatitis B Virus (HBV)

HBV is a small, enveloped, partially single-stranded, circular DNA virus belonging to the *Hepadnaviridae* family. WHO estimates that 296 million people were living with a chronic HBV infection in 2019 [40]. Symptoms of acute HBV infection range from asymptomatic to fulminant hepatitis with cases of ATN [41,42]. Progression to chronic infection is common in infants (95%) and uncommon in adults (5%) [43] except in dialyzed patients (80%) [44], for whom acute infection may be asymptomatic (normal serum transaminase levels). During chronic HBV infection, renal disease will develop in up to 6% of patients [45], sometimes without apparent liver disease. Membranous nephropathy (MN) is the most frequent renal disease and has a poor nephrologic prognosis [46]. Pre-existing immune complexes can deposit on the glomerular basement membrane (GBM) or can assemble within the GBM. Renal improvement has been reported after HBeAg clearance in children with MN, emphasizing its role in pathogenesis [47]. HBV-induced MN may also coexist with anti-PLA2R nephropathy, but the frequency of this co-occurrence remains debated [48]. Compared to patients with idiopathic MN, lower levels of circulating complement and more frequent segmental glomerular damage, mesangial cell proliferation and tubulointerstitial damage on histology were observed in HBV patients [49]. Membranoproliferative glomerulonephritis (MPGN) is frequently reported and is induced by HBsAg-related immune-complex deposits [50]. Patients may also present with a protracted course of HBV-related mixed cryoglobulinemia (MC)—mainly type II—often with severe renal manifestations involving nephrotic-range proteinuria and AKI [51]. Polyarteritis nodosa was historically associated with HBV, but this has dramatically changed due to HBV vaccination over the last several decades [52]. Some authors have suggested that IgA nephropathy [53] and minimal change disease (MCD) [54] may be associated with HBV. Finally, an atypical case of HBV-associated lupus-like glomerulonephritis has been recently reported [55]. HBV binds to highly sulfated HSPGs (heparan sulfate proteoglycans), and glypician 5 mostly presents on the surfaces of hepatocytes and uses sodium taurocholate cotransporting polypeptide (NTCP) to penetrate human cells via endocytosis [56]. This explains the high level of the virus’s hepatocyte tropism and the lack of directly HBV-induced damage observed in the kidney.

Antiviral agent nucleot(s)ide analogues (NAs) with limited nephrotoxicity should be used in cases of renal disease [57]. Adding immunosuppressive drugs, including corticosteroids, may be necessary [58]. Interferon is contraindicated in patients with HBV-associated immune disorders [59]. Finally, HBV vaccination is recommended in at-risk populations, sometimes with reinforced schemes and periodical serological assessments [60,61].

### 3.3. Hepatitis C Virus (HCV)

HCV is an enveloped, single-stranded RNA virus belonging to the *Flaviviridae* family with both hepatotropic and lymphotropic properties [62]. Around 58 million people were infected with it worldwide in 2019 [63]. Acute infection symptoms are similar to those of acute HAV or HBV hepatitis. Chronic HCV infection establishes in ~80–85% of cases following acute infection [64] and is associated with frequent extrahepatic manifestations, notably renal type II and III MC [65] in about 15 to 20% of cases [66]. HCV infection is associated with an increased risk of developing end stage renal disease, especially with HCV genotype 1 [67]. Briefly, HCV also binds to the highly sulfated HSPG present on hepatocytes, followed by a complex process of attachment and internalization involving TIM1, lipoproteins, cluster of differentiation 81 (CD81), scavenger receptor class B type I (SR-BI), claudin 1 (CLDN1) and occludin (OCLN), which prevent penetration and subsequent direct damage in renal cells [68]. The glomerular disease most commonly associated with HCV is MPGN with type-II MC, which was first described in 1993 [9]; see Figure 1b. Johnson et al. reported patients with hypocomplementemia and detectable rheumatoid factors such as cryoglobulins and circulating immune complexes in almost all. MN [69], FSGS [69,70], fibrillary and immunotactoid glomerulopathy [71,72,73] have also been associated with HCV, and to a lesser extent IgA nephropathy [74]. HCV infection in KTRs is an independent risk factor for allograft loss [75] and is associated with proteinuria, chronic rejection, transplant glomerulopathy [76], post-transplant diabetes mellitus [77] and immune-complex glomerulonephritis [78]. However, the recent availability of new potent and well tolerated directly acting antiviral (DAA) treatments against HCV may change renal prognosis [79], as they lead to a very high level of sustained virological control [80,81]. Immunosuppressive therapies, such as steroids, alkylating agents and plasma exchanges, remain potentially necessary when severe or refractory-after-DAA HCV-associated renal disease occurs [79]. Although DAAs have a clear benefit in HCV-associated renal disease, post-treatment monitoring may be useful [82].

### 3.4. Hepatitis E Virus (HEV)

HEV is a nonenveloped single-stranded RNA virus belonging to the *Hepeviridae* family [83]. Four major genotypes (HEV 1-4) with differential geographic repartitions are described: HEVs 1 and 2 are mainly in developing countries due to poor sanitation conditions, and HEVs 3 and 4 are mainly in developed countries as zoonotic diseases [83,84]. Acute HEV, mainly presenting in self-limited, mild, acute hepatitis but fulminant forms, may occur, especially in pregnant women infected with genotype 1 [83]. Extra-hepatic manifestations are common, including as neurological, renal, hematological and autoimmune diseases [85]. HEV infection has been associated with immune-mediated kidney diseases [86] even in immunocompetent patients [87], and AKI occurred in 8.6% in a cohort study [88]. Renal disease in acute HEV infection cases was slightly worse than in HAV infection cases [89]. HEV genotype 1, and HEV genotype 3 especially, have been strongly associated with renal disease [83,90]. A recent study emphasized the roles of immune cells, renal epithelium and the signal axis IFN-/chemokines and IL-18 in renal disease during HEV [91]. The receptor for HEV attachment remains unknown, and candidate receptors are still being identified [92]. Many host factors have been associated with cell penetration by naked and quasi-enveloped HEV, which are both infective virions that can only affect hepatocytes in vivo [93]. To date, there is no evidence of direct nephrotoxicity by the HEV, despite urinary excretion [94].

Glomerulonephritis is the most frequent described disease [85,87] with MPGN with or without cryoglobulinemia [95,96] and MN [97]. IgA nephropathy has also been reported in solid-organ transplant recipients [96]. A higher risk of allograft rejection has been reported following post-transplant HEV infection [86,98]. Ribavirin has antiviral activity on HEV and has been associated with good outcomes [97]. Above all, in recipients of solid-organ transplants, reduction in immunosuppressive therapy appears necessary to obtain viral clearance and may be sufficient in some cases [99].

## 4. Respiratory Viruses

### 4.1. Severe Acute Respiratory Syndrome Coronavirus 2 (SARS-CoV-2)

SARS-CoV-2 is an enveloped, single-stranded RNA virus belonging to the *Coronaviridae* family. It recently emerged as a major threat to KTRs and hemodialysis patients, especially during the first waves of the Coronavirus disease 2019 (COVID-19) pandemic [100]. Moreover, immunosuppressive drugs being prescribed to solid-organ transplant recipients worsens the clinical course [101] and are associated with their poorer post-vaccination humoral responses, despite multiple booster injections [102].

In the specific setting of renal injury related to SARS-CoV-2 infection, several presentations have been described. The frequency of acute kidney injury (AKI) varies among published series, ranging from 5% to 50%, depending primarily on the severity of infection [100,103,104]. In published cohorts, the main cause of AKI was acute tubular necrosis (ATN) [105,106,107,108], which was present in approximately 50% of cases and coincided with multiple factors (hemodynamic changes, rhabdomyolysis, nephrotoxic drugs, etc.) but was possibly a direct viral injury [109]. Evidence of SARS-CoV-2 in the kidney tissue as a direct pathogenic factor has been debated [110]. In a recent review, Hassler et al. found that SARS-CoV-2 was identifiable in up to 50% of kidney biopsies using different techniques [111]. Indeed, angiotensin-converting enzyme 2 (ACE2) is the major cellular receptor for SARS-CoV-2 spike protein, and its expression on proximal tubular cells and podocytes has been demonstrated [112]. Moreover, blockading of ACE2 interaction with SARS-CoV-2 prevented cell infection in an in vitro model of kidney organoids [113]. Thus, SARS-CoV-2 infection may locally promote the production of molecules associated with hypoxic damage and inflammatory pathways, including expression of interferon and complement activation, explaining the histological lesions observed during AKI [114,115,116].

The other major cause of AKI is the development of collapsing FSGS in 25 to 40% of cases of so-called COVID-19 associated nephropathy (COVAN) [117,118,119,120]. The development of COVAN, similarly to HIVAN, appears to be particularly related to the *APOL-1* genotype rather than disease severity, according to the “two-hits” theory [121,122,123,124]; see Figure 1c. Although rare, other histological findings have been described, among which, acute interstitial nephritis [125] and thrombotic microangiopathy [126,127] are the main ones. Finally, allograft rejection lesions have been observed in 50% of for-cause biopsies in KTRs presenting an AKI during or after a SARS-CoV-2 infection [118,119,126,128]. However, the exact causality may be difficult to assess if immunosuppressive drugs have been used previously [129].

### 4.2. Influenza Virus

Influenza virus is an enveloped, single-stranded, fragmented RNA virus belonging to the *Orthomyxoviridae* family. An impact of influenza infection on the kidneys, mostly studied during the H1N1 pandemic, is uncommon but can contribute to a deterioration in the patient’s condition. AKI was observed in approximately 50% of patients hospitalized in intensive care units and was associated with significantly worse outcomes [130]. However, few studies investigated histological injuries in these patients. ATN was the major finding in postmortem renal biopsies of patients infected with influenza A virus [131,132], whereas it may be absent in patients with less severe presentations of the disease [133]. The causes are likely multiple, involving a variety of factors, such as infection-related hypovolemia, drug toxicities and possible aggressive fluid accumulation [134]. In most cases, the development of rhabdomyolysis has been reported as an additional risk factor for AKI in these patients [135,136]. In addition, a few cases of glomerulonephritis [137,138,139] or thrombotic microangiopathy [140] have been described. However, none of these have evidenced influenza’s presence in kidney epithelial and/or glomerular cells. This is probably linked to the virus-specific tropism for respiratory epithelial cells, mainly through fixation of sialic acid by the viral protein HA [141,142].

## 5. The Herpesviridae Family

Eight members of the family Herpesviridae strictly infect humans. They are enveloped, double stranded DNA viruses and are characterized by their ability to persist lifelong in a latent state after primary infection. Mechanisms of herpesviruses’ entry into the cell share the use of a conserved core fusion complex (gH/gL and gB) and some specificities that depend on the herpesvirus involved [143].

### 5.1. Herpes Simplex Viruses (HSV) 1 and 2

Primary HSV infection is very common and usually occurs during childhood; possible recurrences throughout life that may be triggered by several known factors [144]. In immunocompromised patients, HSV infection may result in severe clinical presentations. In addition to the common fusion complex of all herpesviruses, the viral protein gD helps with HSV’s entry into cells. Three receptors of gD have been described: herpes virus entry mediator (HVEM), which is a TNF receptor; nectin-1; and a modified form of heparan sulfate. The expression of these specific gD receptors in human cells explains the tropism of HSV: HVEM is expressed primarily in immune cells, and nectin-1 is mainly found in neurons [145,146]. Treatment with acyclovir is often required in these patients, which can lead to nephrotoxicity, by far the most common cause of AKI in HSV infections. Of note, HSV-related nephropathy by direct or indirect mechanisms appears to be very rare, and only a few case reports of HSV nephritis [147,148,149,150] and glomerulonephritis [151,152] have been reported so far.

### 5.2. Varicella-Zoster Virus (VZV)

VZV’s involvement in nephrology is quite similar to that of HSV: primary infection and recurrence can be more frequent and severe in immunocompromised patients, and specific treatment (also acyclovir) is a source of nephrotoxicity. VZV infection involves the same complex of fusion proteins as other herpesviruses; however, it is more oriented toward cell-to-cell fusion than virus-to-cell fusion [153]. As for HSV, few cases have described the occurrence of nephropathy triggered by VZV infection; they mainly presented as glomerulonephritis [154,155,156,157], but also possibly as nephrotic syndrome [158,159] or thrombotic microangiopathy [160]. It should be noted that none of these studies demonstrated viral antigens in renal biopsies.

### 5.3. Epstein Barr Virus (EBV)

EBV, also known as human herpesvirus 4 (HHV4), infects up to 90% of the population, primarily in childhood, and then establishes a lifelong latent infection in memory B cells [161]. Ephrin receptor A2 is involved in cell infection by EBV through fixation of the gH/gL complex [162]. Moreover, gp42 is a specific EBV protein which recognizes HLA class II, which is mainly expressed by B cells [163]. This particular tropism may explain some aspects of EBV-induced nephropathies, particularly regarding idiopathic nephrotic syndrome (INS) [164]. Indeed, a large French pediatric cohort study recently identified that the detection of EBV DNA in peripheral blood was significantly more frequent in INS children compared to matched controls [165]. One explanatory hypothesis is that B lymphocytes could be self-reactive and produce anti-EBNA-A antibodies that cross-react with certain podocyte proteins [166]. Although this mechanism needs to be proven experimentally, the recent recognition of the B cells’ involvement in INS supports this hypothesis [167]. Another nephropathy induced by abnormal EBV B-cell reactivity is the monoclonal cell infiltrate that may be observed during post-transplant EBV lymphomatous disease, which can lead to AKI [168,169].

Tubulointerstitial nephritis with polyclonal immune infiltrate has also been described repeatedly in the literature as a cause of AKI during EBV infection [170,171]. Few have investigated whether or not the virus is present locally, but those who did failed to prove the presence of direct EBV damage. Thus, the mechanism seems rather related to a disproportionate immune response to EBV infection than to direct toxicity of the virus against tubular cells. In addition, rhabdomyolysis is recurrently associated with EBV-induced AKI and should be identified in these situations [171]. Hemophagocytic syndrome is a rare but dramatic complication of EBV infection caused by a cytokine storm that can lead to severe AKI [172,173]. Finally, some cases of glomerulonephritis and thrombotic microangiopathy have also been reported in the setting of primary EBV infection [171,174,175].

### 5.4. Cytomegalovirus (CMV)

CMV’s seroprevalence among the adult population exceeds 80%, and it has a high geographic variability (>90% in low outcome countries and <50% in some other regions) [176]. It often leads to mild, nonspecific symptoms in immunocompetent patients, but remains a major problem in immunocompromised hosts because CMV primoinfection and/or reactivation can lead to severe organ dysfunction. CMV has a tropism for all cells, but especially epithelial and endothelial cells, for which the entry requires the formation of a viral tetramer (gH/gL with gO) or pentamer (gH/gL with UL128-UL130-UL131A) [177]. The tetramer structure further complexes with PDGFRα, whereas the pentamer structure will complex with neuropilin 2 (NRP2) to permit CMV entry into cells [178,179].

One of the first suspected associations between CMV infection and kidney pathology was the controversial description of CMV-related IgA nephropathy [180]. However, these initial observations were quickly challenged by subsequent studies that found no specific CMV staining using monoclonal antibodies, suggesting that polyclonal antibodies induced non-specific cellular staining [181,182]. This has been recently supported by the absence of CMV DNA in biopsies of patients with IgA nephropathy [183,184,185,186].

Whilst AKI may be frequently associated with CMV replication in KTRs, its causes may be multiple and often not related with a virus-associated nephropathy. Digestive symptoms can be severe in cases of CMV disease and lead to ATN, particularly in KTRs with frail allograft function. Foscarnet, an antiviral drug used in cases of resistance mutations to (val)ganciclovir, is a major source of nephrotoxicity [187].

Besides the virus’s tropism for epithelial cells of the kidney/urinary tract and the virus’s excretion in urine in large quantities for years upon CMV infection, kidney disease is not part of the clinical spectrum of CMV infection, and renal function impairment has only very rarely been reported in neonates and infants. The best-described specific lesion induced by CMV is tubulointerstitial nephritis [188,189,190], which is concordant with the virus’s epithelial urinary tropism [191]. In an Indian cohort, only 10 allograft biopsies out of 2,900 demonstrated CMV nephropathy (including seven for tubulointerstitial nephritis) and the majority of these evolved favorably after antiviral therapy [192]. A more recent American cohort study reported CMV nephritis prevalence of 0.2% and poorer allograft outcomes [193]. Typical cytomegalic inclusion bodies (owl’s eye looking) are rarely observed and are usually very focal; they are most often seen in tubular epithelial cells or endothelial cells and rarely in mononuclear inflammatory cells [194,195,196,197]. Diagnosis can easily be confirmed by immunochemistry or in situ hybridization.

On the contrary, acute glomerulonephritis following CMV infection has been reported, mainly in immunocompromised patients [198], and is supported by in vitro murine models of CMV-induced glomerulonephritis [199]. In kidney transplantation, distinction from allograft rejection may be challenging (due to the rarity of visible cytopathic changes), especially before the occurrence of standardized rejection definitions [200]. Of note, some cases of nephrotic syndrome have also been reported following CMV infection [201,202].

### 5.5. Human Herpesvirus 6 (HHV6)

HHV6 primary infection usually occurs during the first two years of life, leading to non-specific symptoms such as fever and cutaneous rash [203]. CD46 is a major receptor of the HHV6 complex fusion protein and is mainly expressed by T cells [204]. Whilst CD4^+^ T cells are the main target of HHV6, a latent tropism toward kidney epithelial cells has been demonstrated [205]. Reactivation of HHV6 can be observed in immunocompromised patients, remaining predominantly asymptomatic, though some cases of severe encephalitis and colitis have also been described [206,207]. In kidney transplantation, HHV6 reactivation has been associated with CMV reactivation [208], and possibly allograft rejection [209]. To the best of our knowledge, no cases of HHV6 related nephropathy has been reported in the literature to date.

### 5.6. Human Herpesvirus 8 (HHV8)

As for EBV, with which it is closely related, HHV8 is a pro-oncogenic virus known for its involvement in Kaposi sarcoma’s occurrence in HIV patients [210]. Its seroprevalence varies upon geographical localization and is rather low in Europe and North American countries [211]. HHV8 enters into cells through the fixation with the ephrin receptor tyrosine kinase A2 or A4 [212]. Rare cases of HHV8 donor-derived primary infection have been demonstrated following kidney transplantation [213]. Consecutively, in cases of HHV8 replication in the setting of immunosuppression, the risk of Kaposi sarcoma is increased [214,215]. Besides its pro-oncogenic risk, HHV8 replication carries a risk of cytokine storm and hemophagocytic syndrome, which can lead to AKI and may require immunomodulation therapy [216,217]. To date, no case of specific HHV8-related nephropathy has been reported.

## 6. Polyomaviruses 

### 6.1. BK Polyomavirus (BKPyV)

BKPyV, similarly to JC virus, is a nonenveloped, double-stranded DNA virus belonging to the *Polyomaviridae* family. The four classified BKPyV genotypes all lead to mostly asymptomatic infections during childhood. It has a seroprevalence of around 80–90% [218]. BKPyV has a selective tropism for kidney tubular epithelial cells and the urothelium, in which it persists long-life. Caveolar endocytosis of the virus seems to be the main mechanism involved in the entry of BKPyV into kidney tubular epithelial cells [219]. BKPyV infection has become of increasing interest in the last few decades, mainly because of the BKPyV-associated nephropathy (BKPyVAN) occurring as a consequence of a deeply immunocompromised state. BKPyVAN is mainly observed in the setting of kidney transplantation [220,221], even though it has also been described in native kidneys of deeply immunocompromised patients [222]. Following hematopoietic stem cell transplantation (HSCT), urinary BKPyV replication mainly led to hemorrhagic cystitis. Viral reactivation occurs particularly in an immunocompromised state involving T-cell responses [223], but whether the BKPyV source is derived from the recipient or the donor is unclear, even if the donor transmission hypothesis is currently favored. BKPyV is spread cell-to-cell in an ascending way until reaching the tubular epithelial cells. Characteristic cytopathic changes and/or evidence of viral replication by immunochemistry or in situ hybridization in epithelial tubular cells’ nuclei (sometimes restricted to the collecting ducts of the medulla in early cases) comprise the first observable histological pattern of BKPyVAN. These further evolve through tubular-cell necrosis and inflammation, then tubular atrophy and interstitial fibrosis, which have major impacts on allograft function and survival [224,225]. The use of a morphologic BKPyVAN classification into three prognostic classes proposed by the Banff working group is recommended to improve therapeutic management and comparative data analysis [226]; see Figure 1d. Of note, infection of glomerular parietal epithelial cells can also be noticed, and seems to be associated with a worse renal prognosis [7]. Reactional tubular and interstitial local inflammation can also occur, which may challenge the differential diagnosis with acute allograft rejection, especially if viral inclusions are not foreseen. Indeed, even if multiple specific therapies have been evaluated, the reduction of the immunosuppression burden based on the monitoring of viral loads in urine and/or blood is the best treatment to date in order to stabilize the BKPyVAN, which can unfortunately lead to allograft rejection [227,228,229].

Finally, rare glomerular proliferative focal lesions have also been reported, but their significance and correlation with BKPyV are uncertain due to the absence of viral inclusions and their significant impact on allograft function [230].

### 6.2. JC Virus

The JC virus was described the same year as BKPyV in an immunocompromised patient that presented with progressive multifocal leukoencephalopathy [231]. Similarly, it has a tropism for urothelial epithelial cells and can reactivate during an immunocompromised state [232,233]. Lactoseries tetrasaccharide c (LSTc) are the main receptors of the viral protein VP1 permitting JC virus entry into the cell, and type-2 serotonin receptors are required to facilitate entry of the virus [234,235]. Interestingly, LSTc and serotonin receptors are both expressed by kidney cells and in the neurological choroid plexus, concordantly with the clinical described tropism of JC virus [236].

However, contrary to BKPyV, JC virus reactivation usually does not coincide with viral nephropathy, and almost never affects kidney allograft function [237,238]. Of note, some have reported the possibility of JC-virus-related nephropathy; the incidence is estimated to be between 0.2 and 0.4% in allograft biopsies. JC-virus-related nephropathy can occur both early and late following transplantation, unlike BKPyVAN, which typically occurs in the early post-transplant period [239,240,241]. All cases presented polyomavirus cytopathic changes and/or SV40 positive staining with JC virus blood replication, in the absence of BKPyV viremia. Associated histological lesions were mainly represented by non-specific tubulo-interstitial chronic changes and glomerulosclerosis, and casualty with JC virus may be debated due to the late post-transplant time of the biopsies. However, in about 50% of JC-virus-related nephropathies, allograft function seemed to be impacted, and moderate to severe tubulitis was noted. Evolution was globally favorable following immunosuppression reduction.

Surprisingly, a nephroprotective effect of JC viraemia has recently been suggested [242]. The proposed hypothesis is a possible effect on *APOL-1* inhibition (opposite effect to the HIVAN mechanism), but these results may also simply reflect a lower global inflammatory state, which is well known to accelerate chronic kidney disease progression [243,244]. Further studies are required in this field to clarify the precise pathophysiological mechanisms.

## 7. Others 

### 7.1. Dengue Virus (DENV)

DENV is an arthropod-borne, enveloped, single-stranded RNA virus (arbovirus) belonging to the *Flaviviridae* family with four known serotypes (DENV-1, -2, -3, -4) [245]. DENV infection may manifest as dengue fever or occasionally as severe dengue: dengue hemorrhagic fever (DHF) and dengue shock syndrome (DSS). Dengue-associated AKI (DAKI) occurs especially through ATN during DHF and DSS and is mainly caused by hypovolemia secondary to bleeding, hyperpermeability and insufficient fluid intake [246,247,248]. Several cases of rhabdomyolysis with AKI have been reported [249,250,251]. DENV entry pathways into host cells are complex. Briefly, glycosaminoglycans and DC-specific intercellular adhesion molecule 3-grabbing nonintegrin (DC-SIGN) are the main routes for DENV attachment and receptor factors [252]. Direct renal injuries by DENV have been suggested considering studies which detect viral antigens in tubular epithelial cells [253]. However, there are no current data to suggest DENV replicates in renal cells [253], and viral particles may rely on deposition of DENV antigen complexed to antibodies or primarily degraded and/or phagocyted virus [254]. Glomerular injuries have been described in animal models [255] and human studies [256], notably with FSGS [257,258] and MPGN [259], without correlation with the *APOL-1* risk allele but with demonstrated infection in the kidney tissues. Finally, thrombotic microangiopathy has also been reported [260,261]. No antiviral is currently available, but some DENV inhibitors are in the development pipeline [245]. Of note, other arbovirus infections have also been associated with renal damage, mainly ATN and thrombotic microangiopathy [257,262,263].

### 7.2. Hantaviruses 

Hantaviruses are enveloped, tri-segmented, single-stranded RNA viruses belonging to the *Hantaviridae* family. Two distinct clinical syndromes are classically defined—although overlap may exist: hemorrhagic fever with renal syndrome (HFRS) caused by old-world (Europe and Asia) hantaviruses and the more severe hantavirus cardiopulmonary syndrome (HCPS)—with no specific renal disease—caused by new-world (North and South America) hantaviruses [264]. Both diseases present as a sudden-onset flu-like syndrome, followed by rapid clinical and biological deterioration [265]. Humans get infected through exposure to infected rodents or their excreta, especially contaminated dust particles [264]. HFRS caused by hantavirus ranges from subclinical to severe renal and hemorrhagic disease with a classical five-phase evolution [264]. A milder form of HFRS called nephropathia epidemica due to Puumala virus classically provokes initial oliguria followed by marked polyuria [266]. Shortly, hantavirus Gn/Gc glycoproteins allow virus attachment and entry through interaction with β-integrins (β1–3) [264]. A constrained interaction between a hantavirus clade and certain host factor has recently been reported, which may contribute to deciphering the hantaviruses’ specifications [267]. Specific impairment of podocytes, endothelial cells and epithelial cells can be observed via direct diffuse viral infection of glomeruli and tubules [8]. Endothelial cell dysfunction with capillary leakage associated with cytokine storm and CD8^+^ lymphocyte infiltrate leads to hypovolemia and ATN [268]. These pathological processes are, however, rather more functional than lesional, as glomerular histological findings are mostly normal and pathological patterns are mainly represented by interstitial nephritis with dominant lymphocytic inflammation associated with interstitial hemorrhage and acute tubular necrosis [269]. Less often, hantaviruses can provoke nephrotic and nephritic syndrome due to proliferative glomerulonephritis [270]; and may lead to transplant rejection [271], possibly through T-cell activation and the oversized immune response. The treatment is primarily supportive and often requires intensive care unit admission with special attention to fluid balance and an occasional need for dialysis. Some antivirals may provide beneficial effects; however, their impact should be further studied [272]. Rodent control and worldwide epidemiological surveillance are essential to detect emerging cases [273].

### 7.3. Parvovirus B19 (B19V)

B19V is a nonenveloped, single-stranded DNA virus belonging to the *Parvoviridae* family. B19V is a ubiquitous, strictly human pathogenic virus. Its transmission route relies on respiratory droplet projections. B19V is characterized by a tropism for erythroid progenitor cells, but it can persist in many non-erythroid tissues as well [274,275]. Three cell receptors/coreceptors have been identified: the glycosphingolipid globoside (globotetraosylceramide (b4Cer)) that is expressed mainly in the human erythroid progenitor cells and the main virus receptor; the Ku80 autoantigen conferring attachment in different cells; the α5β1 integrin acting as a coreceptor [276]. Most symptoms occur secondarily to immune complex formation [277], explaining partly the differences between immunocompetent and immunocompromised patients. B19V is responsible for an acute febrile cutaneous eruption in children, known as fifth “disease”; and fever, arthralgia and eruption in adults. The main reported complications are erythroblastopenia in patients with hemoglobinopathies, and chronic anemia in immunosuppressed patients—including patients on hemodialysis and renal transplant recipients [278]—with prolonged infection. Renal disease may occur through cytopathic effects or immune complex deposits. MPGN or MCD have been rarely described in patients with B19V infection; there is not clear recognition of a causal link [279,280]. In KTR, clinical symptoms are often mild or absent, low-grade fever being the only common one. There is no specific renal involvement described in transplant recipients. The consequences of B19V infection on kidney allograft rejection risk are still controversial [278]. There is no antiviral therapy for B19V, and polyvalent immunoglobulins are recommended in cases of erythroblastopenia in immunocompromised patients [281], which is associated with reduction of immunosuppression [282].

### 7.4. Human Adenovirus (HAdV)

HAdV are non-enveloped, double-stranded DNA viruses belonging to the Adenoviridae family. They are ubiquitous and easily transmitted through inhalation of aerosolized droplets or the fecal–oral route [283]. More than sixty types divided into seven species from A to G have been described; these have specific characteristics, such as host cell tropism or attachment and entry receptors [284]. HADV-B2 has kidney and urinary-tract tropism and uses CD46 (also called membrane cofactor protein) and desmoglein-2 as attachment receptors to bind the distal knob domain of its fiber protein [285]. HAdV disease usually manifests as self-limited and particularly affects young children. Organ-specific or disseminated disease may be seen in immunocompromised patients, sometimes in severe and lethal forms, especially in HSCT recipients [283,286]. In KTR, HAdV viremia mainly occurs during the first year, especially the first 3 months [287,288], and sometimes presents in asymptomatic forms. Hemorrhagic cystitis is the most frequent disease in KTR, followed by renal allograft dysfunction caused by tubulointerstitial nephritis [289,290], sometimes necrotizing. Although HAdV infection is common, interstitial nephritis is rare; only a few biopsy proven cases have been reported in the literature [291]. A case of interstitial nephritis with a granulomatous compound has also been reported [289]. The differential diagnosis with allograft rejection may thus be challenging, relying on cytopathic modifications [287] (which can overlap BKVPyN) and immunostaining for adenovirus antigens. Reduction of immunosuppression is a mainstay of treatment. Various molecules (cidofovir, brincidofovir, ganciclovir, valganciclovir) exert an antiviral effect on HAdV. However, there is no consensus antiviral therapeutic approach, and strategies are mainly based on case reports [289,292].

### 7.5. Measles Virus (MeV)

MeV is an enveloped, single-stranded RNA virus belonging to the Paramyxoviridae family [293] that affects only humans [294]. MeV is easily transmitted by air and is responsible for high morbidity and mortality, especially in children [295], leading to major costs for health systems [296]. MeV uses the signaling lymphocyte activation molecule (SLAM), CD46 and nectin-4 as cellular receptors which interact with two MeV envelope glycoproteins, hemagglutinin and the fusion protein, to attach and enter human cells [297]. MeV provokes a febrile disease with upper respiratory symptoms and maculopapular exanthema [298]. Kidney injury during measles infection has been rarely reported. In a small series of 38 patients hospitalized for measles, almost half had clearance of less than 90 mL/min, and only one had acute renal failure with proteinuria [299]. No specific MeV antiviral therapy exists. Potent live attenuated vaccines exist and are highly efficient, but the eradication of the disease is challenged by vaccine hesitancy and immunization program failures [300].

## 8. Conclusions

The occurrence of virus-associated nephropathy depends on multiple factors: the local epidemiology of the virus, and its ability to infect renal cells or to lead to immune-mediated renal damage influenced by the patient’s underlying immune response (Table 1). Current knowledge provides the physician with important guidelines for diagnosing episodes of AKI associated with a viral infection and may help inform the decision of whether a renal biopsy should be done. Even if no clear consensus exists for the majority of cases, the therapeutic decision may be supported by the underlying mechanism of the viral nephropathy (mainly direct viral toxicity or indirect, immune-mediated).

## Figures and Tables

**Figure 1 ijms-23-12014-f001:**
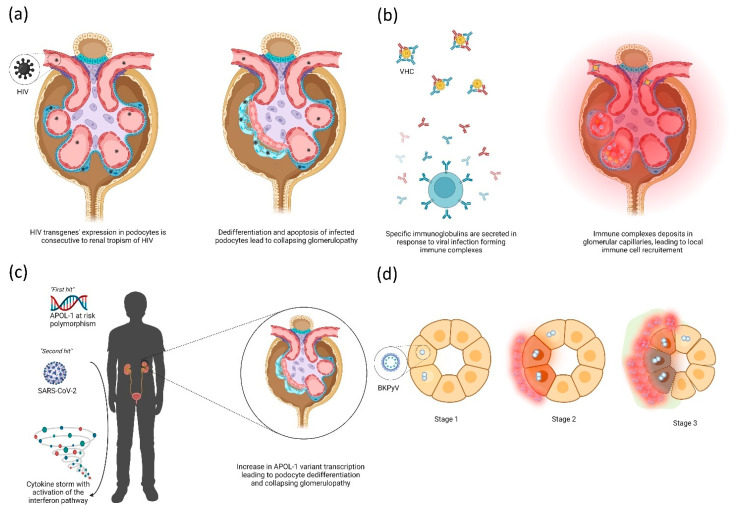
Main representative direct (**a**,**d**) and indirect (**b**,**c**) mechanisms observed in viral nephropathies. (**a**) Podocyte infection by HIV leads to expression of HIV transgenes which promote dedifferentiation and vacuolization of podocytes. Abnormal podocytes detach from the glomerular basement membrane, leading to collapse of the capillary coves and flocculus. (**b**) The B cell-mediated reaction to viral infection, especially during HCV, can lead to the production of immune complexes consisting of specific IgG and IgM that fix viral particles, possibly with a cryoglobulinemic property. These complexes will further disseminate through the whole organism, and notably throughout the glomerular capillaries where they induce neutrophil and T cell recruitment, leading to the MPGN pattern. (**c**) Patients with the APOL-1 at-risk polymorphism carry a risk of collapsing glomerulopathy, especially if a “second” hit occurs involving the interferon pathway. The APOL-1 variant transcript is thus increased, leading to podocyte differentiation and progression to collapsing glomerulopathy. (**d**) Epithelial renal tubular cells are infected with BKPyV, leading to intranuclear viral inclusions (Stage 1), further evolving towards cytoplasmic modifications with a local immune reaction (Stage 2) and finally cell apoptosis and fibrosis (Stage 3).

**Table 1 ijms-23-12014-t001:** General overview of histological patterns depicted in viral nephropathies. “+/−“ refers to several cases reported, “+” refers to a rarely described association and “++” refers to a commonly observed association.

	Tubulo-Interstitial			Vascular	Glomerular			
	*ATIN*	*ATN*	*Myoglobin tubulopathy*	*TMA*	*MPGN*	*MN*	*MCD*	*cFSGS*
HIV	+			+/−	++			++
HAV	+/−	+			+/−			
HBV		+/−			++	++	+/−	
HCV					++	+		
HEV					+	+		
SARS-CoV-2	+/−	++		+/−	+/−			++
Influenza virus		++	+	+/−	+/−			
HSV	+/−				+/−			
VZV				+/−	+/−		+/−	
EBV	+		+	+/−	+/−		++	
CMV	+	++			+/−			
HHV6								
HHV8								
BKV	++				+/−			
JCV	+/−							
DENV		++	+	+/−	+/−			+/−
Hantavirus		++						
B19V					+/−		+/−	
HAdV	+							
MeV								

*ATIN*: acute tubulo interstitial nephritis. *ATN*: acute tubular necrosis. *TMA*: thrombotic microangiopathy. *MPGN*: membranoproliferative glomerulonephritis. *MN*: membranous nephropathy. *MCD*: minimal change disease. *cFSGS*: collapsing focal segmental glomerulosclerosis.

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
