# Peer review of "Virus-Associated Nephropathies: A Narrative Review"

_ijms, 2022, doi:10.3390/ijms231912014_

Round 1

Reviewer 1 Report

Comments: Manuscript review is very promising and smart work. Few issues should to be studied to clarify such this interesting work.

1-    Few English Typos errors should to be revised  thoroughly over all the text

2-    Introduction was not completely supported by REFs. Authors should to add new REfs for introduction paragraphs

3-    The paragraphs from lines 79 to 86, 288 to 289  and 439 to 440 should be rearranged as other text.

4-    The mechanism by which virus can entry into cells was not addressed. Authors should to describe which protein is responsible for virus entry in different organs (liver, kidney, lung)

5-    The drug used to treat virus infection should have been added as subsection. Authors have to add the  drugs that had been used to treat virus in different infections

6-    Anatomy structure of virus should have been obtained. Authors also should to address if there is difference in  spike glycoprotein

7-    Authors should to write comparison in symptoms, diagnosis and cytokines between Hepatitis, Respiratory Herpesviridae and Polyomaviruses and to be summarized in table.

8-     Authors should to explain why some drugs are not succeed to treat virus.

9-    The stage of virus infection was not explain in patients with  nephropathies. Authors should to write in deep how virus can penetrate into cytoplasm and when will be duplicated with DNA in host cells.

10-  REFs should be revised according to format style of IJMS -MDPI journal .

11. Scheme to illustrate how virus can affect kidney in patents  with native or transplanted kidneys

Author Response

Reviewer 1

Few English Typos errors should to be revised  thoroughly over all the text.

Thanks for this remark. The manuscript have been provided by another English revision.

Introduction was not completely supported by REFs. Authors should to add new REfs for introduction paragraphs

Thanks for this point. We added references in the introduction section as suggested.

The paragraphs from lines 79 to 86, 288 to 289  and 439 to 440 should be rearranged as other text.

Thank you for this remark. The typo have been modified so.

The mechanism by which virus can entry into cells was not addressed. Authors should to describe which protein is responsible for virus entry in different organs (liver, kidney, lung)

Thank you very much for this pertinent remark. We added some pathophysiological precisions for each virus to the final manuscript. However, we decided not to detail thorougly this section considering the word restrictions to be observed. Besides we tried to add only the most relevant data on this subject in view of the scope of this review.

The drug used to treat virus infection should have been added as subsection. Authors have to add the  drugs that had been used to treat virus in different infections

Thank you for this point. We completed the therapeutic paragraph in each virus section.

Anatomy structure of virus should have been obtained. Authors also should to address if there is difference in  spike glycoprotein

Thank you for this remark. A brief description of the virus main component have been provided in the beginning of each section. Besides, I’m not sure to understand your point about spike glycoprotein (was it a specific question for SARS-Cov-2 ?)

Authors should to write comparison in symptoms, diagnosis and cytokines between Hepatitis, Respiratory Herpesviridae and Polyomaviruses and to be summarized in table.

The aim of this review is to provide a summary and clear description of virus associated nephropathies. Thus, we deliberately not deepen general symptoms non related to kidney – as if we do so for each virus, this will significantly longer the manuscript. Moreover, we already had a point from another reviewer in order to concise our purpose on kidney and avoid generalities which our readers are probably already aware of.

Authors should to explain why some drugs are not succeed to treat virus.

As described above, we increased the therapeutic section for each virus to give more details on this point.

The stage of virus infection was not explain in patients with  nephropathies. Authors should to write in deep how virus can penetrate into cytoplasm and when will be duplicated with DNA in host cells.

As described above, we provided some precisions on pathophysiological mechanisms involved in kidney nephropathies regarded to your point. We described more precisely the way of entry but without detailing what happens within the cell. These specific data didn’t appear mandatory for the aim of this review.

REFs should be revised according to format style of IJMS -MDPI journal .

As you recommended, we modified the style of our references according to the American Society of Chemistry (MDPI style)

Scheme to illustrate how virus can affect kidney in patents  with native or transplanted kidneys

 We finally opted not to specifically distinguish native and transplant kidney injuries – and changed the title of our review in this way- but to give an overview of each viral’s kidney disease. Accordingly, Figure 1 summarizes the main mechanisms involved during virus-associated nephropathies. 

Reviewer 2 Report

Authors provided nice and comprehensive review about influence of different viral infections on kidneys and their pathophysiology. Still, majority of the paper discuss about influence of viral infection on healthy kidney, with only minor (two for HAV, one of HBV,... or even zero sentence per virus - HIV, HSV, VZV etc) mentioning of how that virus acts in organ transplant recipients. If so, I suggest two options: either removing the "transplated kidneys" from the title or giving more space for that part in the text, discussing more deep and more precise about specificity of that viral infection in transplated kidney, how it different form healthy kidney reaction to that viral infection etc. 

Also, according to my opinion, introduction in each virus is too long with a too wide and already very well known facts about each virus. It would better to focus on influence of that virus on kidneys, because your audience is familiar with the etiology, epidemiology and typical clinical presentation of each viral infection, and those fact do not deserve so much space in a review article named: Virus-associated nephropathies...  

Author Response

Authors provided nice and comprehensive review about influence of different viral infections on kidneys and their pathophysiology. Still, majority of the paper discuss about influence of viral infection on healthy kidney, with only minor (two for HAV, one of HBV,... or even zero sentence per virus - HIV, HSV, VZV etc) mentioning of how that virus acts in organ transplant recipients. If so, I suggest two options: either removing the "transplated kidneys" from the title or giving more space for that part in the text, discussing more deep and more precise about specificity of that viral infection in transplated kidney, how it different form healthy kidney reaction to that viral infection etc. 

Thank you for this relevant review of our work. We acknowledge that native kidney takes up the largest place on the manuscript compared to transplanted kidneys. This is linked to the fact that virus associated nephropathies have been mainly described in native kidneys (as HIVAN, COVAN, VHC-cryoblobulinemia, VHB-Membranous nephropathy etc). Indeed, the most specific virus associated nephropathies in transplanted kidney are the one involving Polyomaviruses. Causes for these differences may be multifactorial : kidney allografts are less frequent than native kidney in the global population, and rare events are consecutively exceptional in these patients. Morevoer, the possible impact of immunosuppression in the prevention of viral associated nephropathies driven by immune mechanisms is unknown.

Thus, in accord to your point, we decided to change the title and talk globally about “virus associated nephropathies” without specificaly differenciate native and transplanted kidneys.

Also, according to my opinion, introduction in each virus is too long with a too wide and already very well known facts about each virus. It would better to focus on influence of that virus on kidneys, because your audience is familiar with the etiology, epidemiology and typical clinical presentation of each viral infection, and those fact do not deserve so much space in a review article named: Virus-associated nephropathies...  

Thank you for this point. Consequently, we reduced the part related to viral generalities and focused on pathophysiological pathways involved in virus associated nephropathies, notably on the cell entry mechanisms.

Round 2

Reviewer 1 Report

Comments of reviewer was revised point by point  and manuscript is more  acceptable